# Accelerometry as a method for external workload monitoring in invasion team sports. A systematic review

**Carlos D. Gómez-Carmona**[1]*, **Alejandro Bastida-Castillo**[2,3], **Sergio J. Ibáñez**[1], **José Pino-Ortega**[2]

**1** Training Optimization and Sports Performance Research Group (GOERD), Didactics of Music, Plastic and Body Expression Department, University of Extremadura, Caceres, Spain, **2** Department of Physical Activity and Sports, International Excellence Campus "Mare Nostrum", Faculty of Sport Sciences, University of Murcia, San Javier, Spain, **3** University Isabel I, Burgos, Spain

* cdgomezcarmona@unex.es

**Data Availability Statement:** All relevant data are within the paper and its Supporting Information files.

## Abstract

Accelerometry is a recent method used to quantify workload in team sports. A rapidly increasing number of studies supports the practical implementation of accelerometry monitoring to regulate and optimize training schemes. Therefore, the purposes of this study were: (1) to reflect the current state of knowledge about accelerometry as a method of workload monitoring in invasion team sports according to the Preferred Reporting Items for Systematic Reviews and Meta-analyses (PRISMA) guidelines, and (2) to conclude recommendations for application and scientific investigations. The Web of Science, PubMed and Scopus databases were searched for relevant published studies according to the following keywords: "accelerometry" or "accelerometer" or "microtechnology" or "inertial devices", and "load" or "workload", and "sport". Of the 1383 studies initially identified, 118 were selected for a full review. The main results indicate that the most frequent findings were (i) devices' body location: scapulae; (b) devices brand: Catapult Sports; (iii) variables: PlayerLoad™ and its variations; (iv) sports: rugby, Australian football, soccer and basketball; (v) sex: male; (vi) competition level: professional and elite; and (vii) context: separate training or competition. A great number of variables and devices from various companies make the comparability between findings difficult; unification is required. Although the most common location is at scapulae because of its optimal signal reception for time-motion analysis, new methods for multi-location skills and locomotion assessment without losing tracking accuracy should be developed.

## Introduction

Workload quantification is defined as the process of recording training and competition workload demands to regulate training volumes and intensities in athletes and to decrease the risk of injuries and overtraining [1]. These demands should not only be assessed overall but also individually as each player will respond differently to the same training workloads [2, 3].

**Funding:** The first author of the present study is a beneficiary of a grant from the Spanish Ministry of Science, Innovation and Universities (FPU17/00407). This study has been partially subsidized by the Aid for Research Groups (GR18170) from the Regional Government of Extremadura (Department of Employment, Companies and Innovation), with a contribution from the European Union through the European Funds for Regional Development. The funders had no role in study design, data collection and analysis, decision to publish, or preparation of the manuscript.

**Competing interests:** The authors have declared that no competing interests exist

Concerning workload quantification, sport science research differentiates between internal and external workload [4]. The internal workload is defined as the biological reaction of the athlete's organism, both physiological and psychological, as a consequence of the external workload performed during exercise and it is measured through different variables like heart rate telemetry, blood lactate, oxygen consumption or rating of perceived exertion (RPE) [5]. In contrast, the external workload is defined as the mechanical and locomotor actions performed by an athlete, measured through various variables like power, speed, changes of speed, changes of direction or impacts [6]. Therefore, current literature suggests adopting strategies for quantifying and monitoring internal and external workload can enable team staff to assess fatigue and fitness level of players in real-time throughout the season [4, 7, 8].

At high levels in sports performance, coaches and sports scientists are constantly trying to find new ways for measuring athletes' performance to obtain an advantage over their opponents [9]. However, training and competition activity and the developments of performance are extremely difficult to measure directly [10]. For this reason, sports professionals have found different methods for measuring the players' workloads indirectly such as inertial measurement units (IMUs) for recording in a reliable and valid way compared to other instruments considered as "gold standard" or "criterion measures" [11–13]. These instruments or diagnostic tests are considered the best available and most accurate under reasonable conditions (e.g. the gold standard for players tracking is video analysis but indirect methods can detect it with high accuracy as Global Navigation Satellite Systems, GNSS, Local Position Measurements, LPM or accelerometry).

In this sense, technological advances have allowed the development of different devices to obtain objective data in indoor and outdoor sports. Since 2001, the Australian Centre of Microtechnological Research through Project 2.5 "Technology of Communication to Athletes Monitoring" has been designing a unique and non-intrusive device for sports monitoring in real-time [14]. These devices are able to record external workload demands such as (a) total distance, (b) work zones concerning velocity or changes of speed, or (c) impacts performed by the athletes [15]. The incorporation of tri-axial accelerometers into these units has provided the opportunity to analyze new load parameters such as three axes acceleration recorded during sports movements, measured in arbitrary units (a.u.) [16].

The validity of accumulated accelerometry-based workload in the three planes of movement has been compared with other internal workload variables such as session RPE (sRPE) or the Edwards method, finding high correlations among indexes [17], and also with muscle oxygen saturation [18] or maximal oxygen uptake [19]. Previous research has also found satisfactory reliability results both in accelerometry-based workload [20, 21]. However, the workload recorded by accelerometers could be affected by the individualized profile of gait biomechanics or the speed of the athlete's locomotion [22]. Nonetheless, accelerometry-based indexes have been used for workload monitoring in invasion team sports [23] such as netball [24, 25], soccer [17, 26], basketball [27–30] and Australian football [31, 32], among others. Carey et al. [31] mentioned in a recent investigation that a multi-variable analysis should be carried out, where accelerometry-based indexes are incorporated with other external and internal workload indicators such as total distance covered, sRPE or high-intensity locomotion.

Since its appearance, the use of accelerometry as a method of workload monitoring has developed greatly. Although accelerometers do not provide information about static actions when an effort is performed without an acceleration (e.g. screenings or a prolonged stance position), their reliability, precision and sensitivity are greater compared to other automatic and semiautomatic time-motion analysis (TMA) technologies such as video-tracking, GNSS or LPM [26, 30]. Automatic and semiautomatic TMA may underestimate the workload

demands because high-intensity actions where there is no locomotion (jumps, collisions, etc.) are classified in the group of low-intensity actions [26]. For these reasons, recent investigations identified that microtechnologies (e.g. wearable microsensors and accelerometers) may represent a valid and practical alternative to TMA and offer distinct advantages compared with TMA such as the relative simplification to analyze data using either proprietary or used-defined algorithms that quantify movement [30, 33]. Given this background, the purposes of the present study were to reflect the current state of knowledge, outline best practices and conclude recommendations about the use of accelerometry as a method of workload monitoring in invasion team sports.

## Methods

### Study design and search strategy

This manuscript is a systematic review [34] about peer-reviewed, scientific papers related to workload monitoring via accelerometry in sports. The Web of Science (Web of Science Core Collection, MEDLINE, Current Contents Connect, Derwent Innovations Index, KCI-Korean Journal Database, Russian Science Citation Index and Scielo Citation Index), PubMed electronics and Scopus electronic databases were searched on 1$^{st}$ May 2020 for relevant articles published between 1$^{st}$ January 2010 and 30$^{th}$ April 2020 using the keywords "accelerometer" or "accelerometry" or "microtechnology" or "inertial devices", and "load" or "workload", and "sport". Reference lists of included articles were scanned to identify relevant studies. Any disagreements were resolved by consensus between two investigators and arbitration by a third investigator.

One investigator conducted electronic searches, identified relevant studies, and extracted data in an unblended, standardized manner. The database search was limited to peer-reviewed journal articles published in English. A systematic review of the available literature was undertaken in accordance with the Preferred Reporting Items for Systematic Reviews and Meta-Analyses (PRISMA) guidelines [35] (Fig 1).

In the present review the inclusion criteria for these articles were: (1) cross-sectional and longitudinal studies written in English, (2) participants were healthy players irrespective of competition level (amateur, well-trained, professional, elite, junior, senior) and sex (male and female), and (3) about invasion team sports following the classification of Read and Edwards [36] divided into three sports modalities: (a) goal throwing games (netball, basketball, handball and lacrosse), (b) try-scoring games (rugby, rugby union, Australian football and American football) and (c) goal striking games (hockey and soccer). Analysis during training or competition was not selected as an exclusion criterion. All included studies were deemed to have suitable ethical approval by a relevant review board.

Studies were excluded if (1) the type of document was case studies, doctoral thesis, books or book chapters, congress communications, patents or reviews, (2) they involved animals, (3) the workload monitoring was performed without accelerometry-based indexes, (4) the study context was outside competitive sports, and (5) they only assessed the reliability and validity of accelerometer raw data or accelerometry-based indexes.

### Data extraction and analyzed variables

The Cochrane Consumers and Communication Review Group's data extraction protocol [35] was used to extract the following information about studies that monitored external workload by accelerometry-based indexes in invasion team sports: (1) authors and date, (2) participant data (including sex and sample size), (3) description of the sport and competition level, ((4) type of session or sport context (training, competition, or both), (5) device and body location,

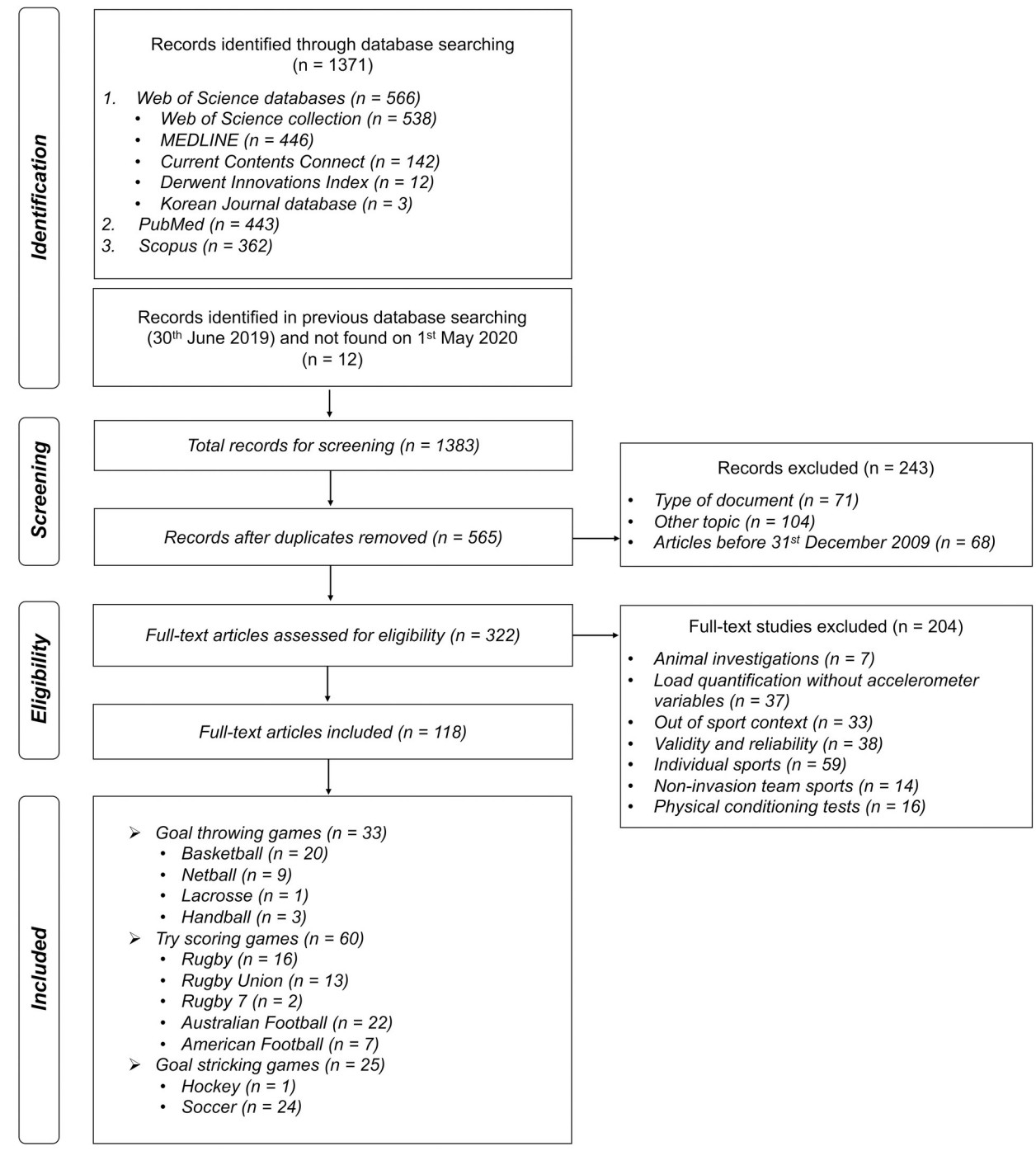

**Fig 1. PRISMA flow diagram displaying the identification, screening, and selection of relevant studies in this systematic review.**

(6) accelerometry-based indexes, (7) technical features of accelerometers (sample frequency, number of accelerometers 3D vs 2D, output range, and previous validity or reliability results), (8) main results and (9) referential values. This process was developed and tested with 10 randomly selected studies. First, one researcher extracted the data from the included studies and a second researcher then checked the extracted data. Disagreements were resolved by consensus.

## Quality of the studies

The quality of the studies was evaluated with a risk-of-bias quality form used for quantitative studies developed by Law et al. [37] (S4 Table) and composed of 16 items in an evaluation process performed by five university full professors with a PhD in sport science and a large number of publications in the field of technology to monitoring sports performance in team sports. Cohen's Kappa was calculated with 95% confidence interval to evaluate the inter-coders reliability and interpreted as: <0.20 *poor*, 0.21–0.40 *fair*, 0.41–0.60 *moderate*, 0.61–0.80 *good*, >0.80 *very good* [38].

Articles were assessed based on purpose (item 1), relevance of background literature (item 2), appropriateness of study design (item 3), sample studied (items 4 and 5), use of informed consent procedure (item 6), outcome measures (items 7 and 8), method description (item 9), significance of results (item 10), analysis (item 11), practical importance (item 12), description of dropouts (item 13), conclusions (item 14), practical implications (item 15), and limitations (item 16). All 16 quality criteria were rated on a binary scale (0/1), wherein two of those criteria (items 6 and 13) presented the option: 'If not applicable, assume N/A'. The introduction of this option for items 6 'Was informed consent obtained?' and 13 'Were any dropouts reported?' was included because, in some studies, the investigators were not required to obtain informed consent (item 6) or report dropouts (item 13). The introduction of the option 'not applicable' allowed an appropriate score for the article, eliminating the negative effect of assuming the value '0' on a binary scale, when in fact that specific item did not apply to that study. For this, the sum of the score of all items was divided by the number of relevant scored items for that specific research design. All articles were classified as (1) low methodological quality (<50%); (2) good methodological quality (51–75%), and (3) excellent methodological quality (>75%).

# Results

## Search, selection and inclusion of publications

1371 articles were identified from the Web of Science ($n$ = 566), PubMed ($n$ = 443) and Scopus ($n$ = 443) database search. In addition, 12 articles identified and selected in previous database searching (30th June 2019) and not found on 1st May 2020 were included, being a total of 1383 articles. These studies were then exported to reference manager software (Zotero), and any duplicates (818 articles) were eliminated automatically. From the remaining 565 articles, 243 did not fulfill the inclusion criteria and were removed after revision of the abstract and another 204 after full-text assessment. At the end of the screening procedure, 118 articles remained for the systematic review related to the invasion team sports modality: (a) goal striking games (soccer and hockey; $n$ = 25) [6, 26, 39–61] (S1 Table), (b) goal throwing games (basketball, netball, lacrosse and handball; $n$ = 33) (S2 Table) [8, 24, 25, 27–29, 62–88] and (c) try-scoring games (rugby, rugby union, rugby seven, Australian and American football; $n$ = 60) [32, 89–147] (S3 Table).

The main reasons for exclusion were individual sports ($n$ = 59), reliability and validity analyses of raw data and workload indexes through accelerometry ($n$ = 38), monitoring external workload without accelerometry-based indexes ($n$ = 37) and non-competitive sports contexts ($n$ = 33). Other reasons for exclusion included studies that analyzed physical conditioning tests ($n$ = 16) and non-invasion team sports ($n$ = 14).

## Quality of the studies

To analyze the quality of the selected studies, the classification designed by Law et al. [37] that is shown as S4 Table was utilized. Previously to quality assessment, an inter-coder reliability

analysis was performed, obtaining a value of 0.93 that represents a very good agreement between observers (Confidence interval 95%: 0.89 to 0.96). The main results of the quality indicators for the selected studies were as follows: (1) the average methodological quality score was 82.3%; (2) Two articles reached the maximum score of 100%; (3) no study obtained a score below 50%; (4) 33 studies obtained a score between 50% and 75% (good methodological quality); and (5), 83 articles reached a rating of >75% (excellent methodological quality).

Four items were mainly related with methodological deficiencies in the selected studies: (1) Criterion 5 where 84.6% of studies did not show an explicit justification of the study sample size; (2) Criterion 16 where 60.7% of articles did not clearly acknowledge the limitations of the study; (3) Criterion 8 where 66.9% did not report the validity of the accelerometry-based index of the device; and (4) Criterion 7 where 42.4% did not report the reliability of the device for accelerometry-based index measurement.

## Scientific journals, sports context, competition level, sex and publication years

Fig 2 shows the scientific journals, sports context, sports level, sex and publication years of the selected studies that use accelerometry-based indexes for workload monitoring in invasion team sports. The trends of topic publications are shown in Fig 2A, where there exists an increasing number of publications with an exponential evolution from 2015. The 118 papers included in the systematic review were published in 27 different journals, with 59.3% appearing in 4 journals, each publishing at least ten articles (Fig 2B).

Most of the studies analyzed competition (58 articles, 49%) or training (32 articles, 27%) separately, with only 24% of selected studies that analyzed both contexts (Fig 2C). Most of the reviewed papers analyzed elite (46 articles, 39%) and professional-level (37 articles, 31%) athletes, although studies in other levels as junior (19%), university (8%) and amateur (2%), and referees (1%) were also carried out (Fig 2D). Finally, 86% of selected studies were performed with males compared to 14% with females (Fig 2E).

## Body location, devices and invasion team sports analyzed

Fig 3 shows the body location, devices and companies, indexes and invasion team sports analyzed by scientific studies through accelerometry-based workload indexes. The most common body location for evaluation was the scapulae (95.8%, 113/118), through MinimaxX (23.7%, 28/118) and Optimeye (33.9%, 40/118) devices in their different versions, developed by the Australian company Catapult Sports (64.4%, 76/118). Most papers were descriptive and assessed maximum accelerations (impacts, collisions) (21/118, 17,8%) or accumulated workload accelerometry-based indexes expressed as arbitrary units (a.u.) through different indexes related to the developer company of the device (16 indexes). Australian football (18.6%, 22/118), soccer (20.3%, 24/118), rugby (13.6%, 16/118) and basketball (16.9%, 20/118) were the most frequently investigated invasion team sports. In addition, impact and collisions were mostly assessed in try-scoring games, and Dynamic Stress Load, Locomotion efficiency, Impulse Load and $PL_{RE}$ in goal striking games (see Table 1 for definitions).

## Accelerometry technical features and based workload indexes

About technical features, 102 articles (86.4%) mentioned the sampling frequency of the accelerometers, being in all cases of 100 Hz, and the triaxial properties of the accelerometers (89.8%, 106/118). Instead, only 9 studies showed the number of accelerometers that composed the devices (4 accelerometers, 6.8%, 8/118; 3 accelerometers, 1/118). Respect to the reproducibility and the accuracy of the accelerometers, 41 articles not reported both aspects (34.75%),

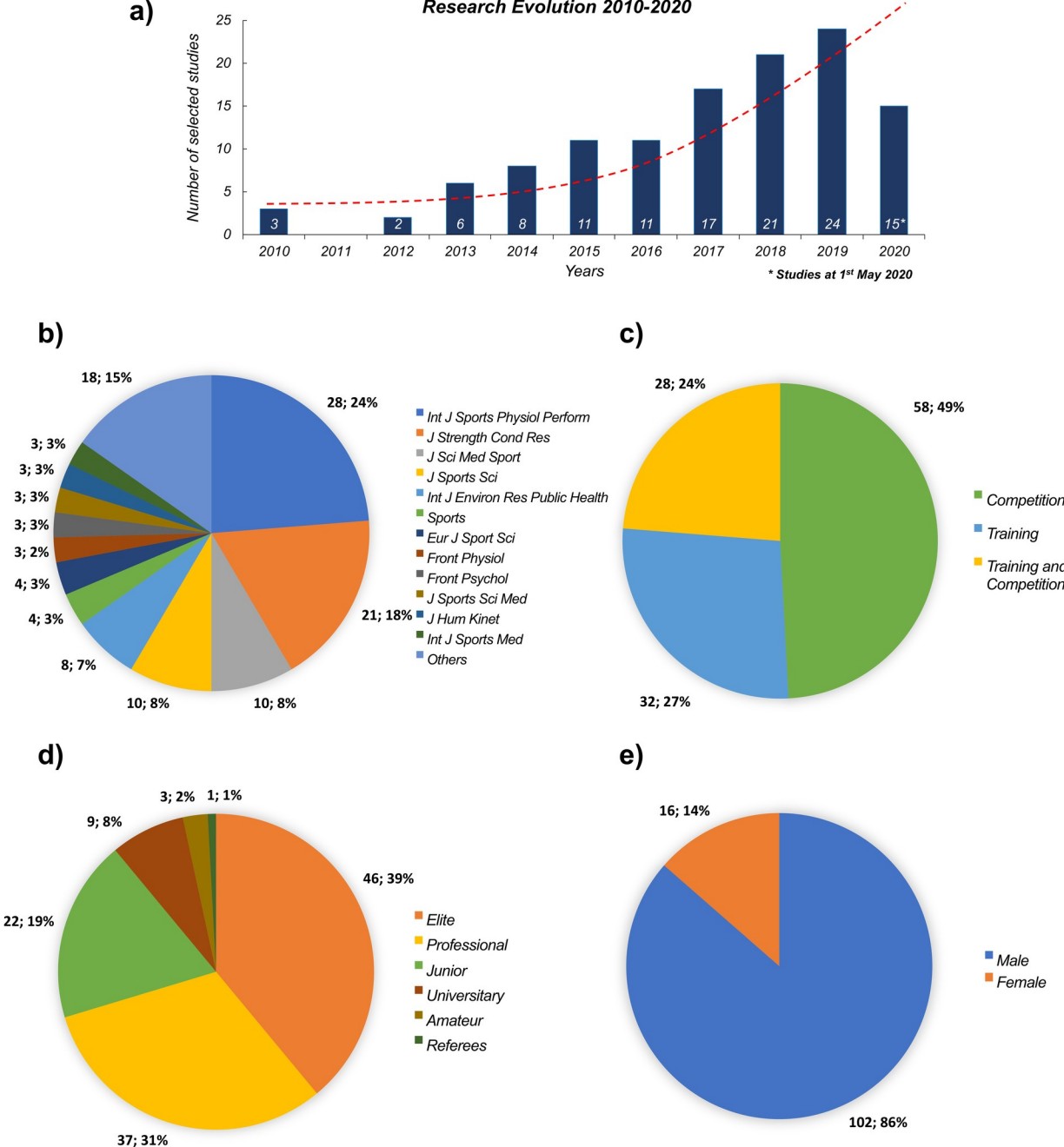

**Fig 2.** (a) Research evolution, (b) scientific journals, (c) type of session, (d) competition level and (e) sex of participants in the selected studies that use accelerometry-based indexes for monitoring workload in sport.

37 articles reported the reliability (31.4%) and 10 reported the validity to measure the accelerometry-based workload index (8.5%). Only 30 articles reported the validity and reliability of the accelerometers that composed the inertial devices (25.4%).

Finally, Table 1 shows the accelerometry-based indexes utilized for workload monitoring with the developer company, description, measurement unit and formula for its calculation. The most frequently used is PlayerLoad™ (PL™) developed by Catapult Sports (77 studies,

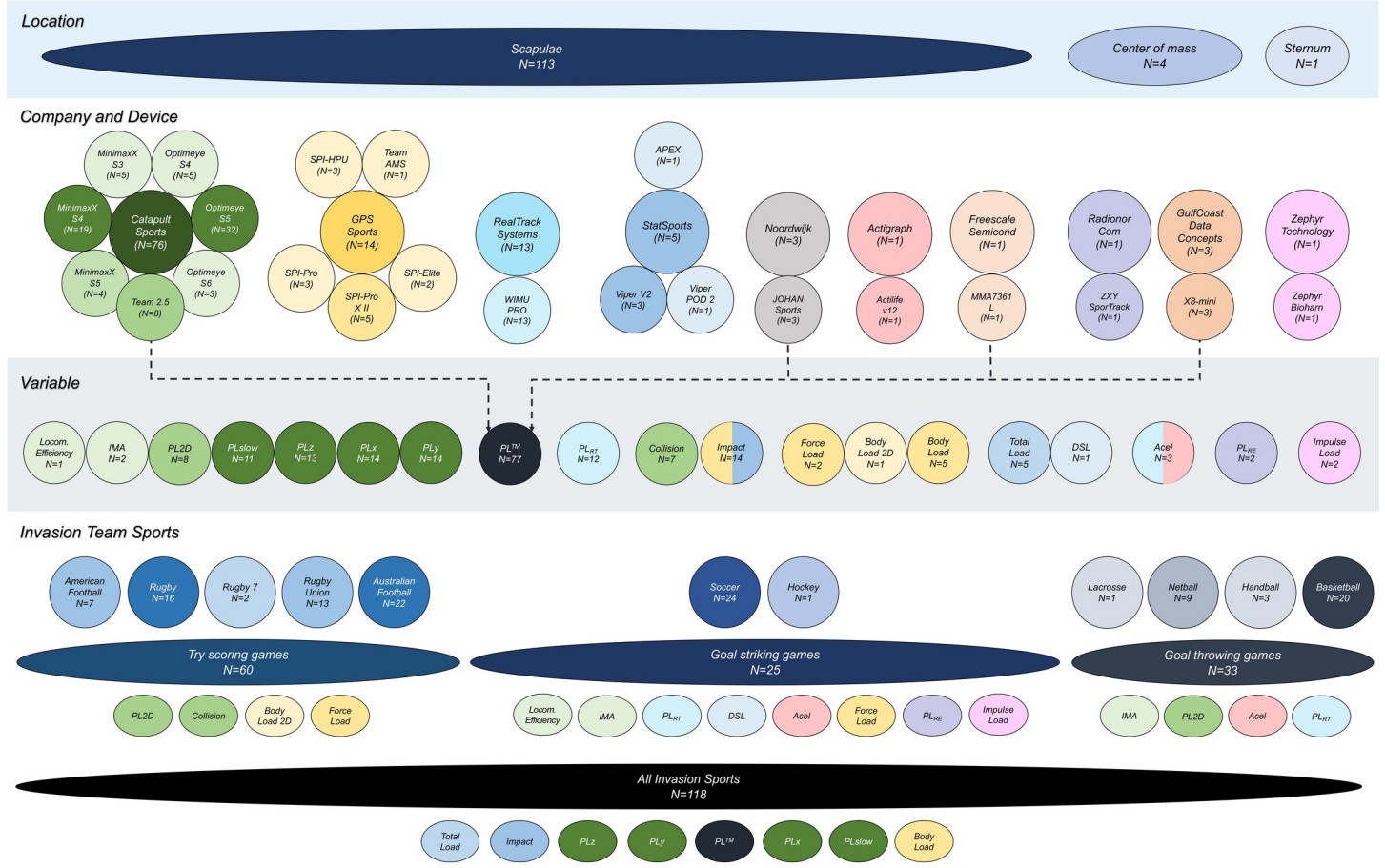

**Fig 3. Classification of selected studies related to body location, device model and company, accelerometry-based workload indexes and invasion team sport.**

65.3%). Also, variations of the original formula such as accelerometry workload at low intensity (PLslow, 11 studies, 9.3%) and divided by axis such as PLx (14 studies, 11.9%), PLz (13 studies, 11.1%) and PLy (14 studies, 11.9%) were utilized (more details in Fig 3).

## Discussion

This manuscript showed a general overview of the use of accelerometry as a method of workload monitoring in invasion team sports, including research evolution, journals, sport modalities and contexts, competition level, sex, device location, accelerometry-based variables and technical features. For this purpose, a systematic review was carried out of the articles related to the study topic [34, 35]. The main results show a rapidly increasing number of publications about accelerometry-based workload monitoring, where training and competition were analyzed separately, in elite and professional-level men, placing the device at the scapulae level and using the PL$^{TM}$ index in invasion team sports in outdoor and indoor conditions.

### Competition vs. training

Most studies analyzed training and competition contexts separately (78%) so that only a limited number compared both contexts (22%). The interrelation between training and competition during sports seasons is essential to achieve the appropriate adaptations, maintain

**Table 1. Accelerometry-based external workload variables utilized in the selected studies in this systematic review.**

| Index | Description | Units | Developing company | Formula |
|---|---|---|---|---|
| AcelT | Root square of the sum of the accelerations in the three axes of movement. | Meters per second square (m·s$^2$) | None | $\sqrt{(x^2 + y^2 + z^2)}$ |
| Body Load (BL) | Accelerometry-load based index in the three axes of movement. Following steps are repeated for each acceleration value: (1) Initialize the Body Load count to 0; (2) Root square of the sum of the accelerations in the three axes of movement (x, y and z); (3) Normalize the magnitude vector by subtracting a notional 1G; (4) If the normalized value is less than 0.25G then go to step 2; (5) Calculate the unscaled Body Load (USBL) contribution for this acceleration vector; (6) Calculate the scaled Body load (SBL) considering the accelerometer logging rate (100 Hz) and Exercise Factor (EF); (7) Calculate the total Body Load as the accumulation of the scaled Body Load count. | Arbitrary units (A.U.) | GPS Sports | 1. BL = 0 <br> 2. $\sqrt{(ay^2 + ax^2 + az^2)}$ <br> 3. NV = V– 1.0 G <br> 5. USBL = NV + (NV)$^3$ <br> 6. SBL = USBL / 100 / EF <br> 7. BL + SBL |
| Body Load 2D (BL2D) | Accelerometry-load based index in the two planes of movement (anteroposterior and mediolateral). Same steps as BL but not considering z-axis in the formula. | Arbitrary units (A.U.) | GPS Sports | 2. $\sqrt{(ay^2 + ax^2)}$ |
| Collisions | For a collision to be detected, the unit was required to be in a nonvertical position, meaning the player was leaning forwards, backwards, or to the left or right. The instantaneous player load was calculated from the sum of the three axes of acceleration. A spike in the instantaneous player load shortly before the change in orientation of the unit was also required for the collision to be detected. | Count (n) | Catapult | Not applicable |
| Dynamic Stress Load (DSL) | It was calculated as the total of the weighted impacts. Impacts were weighted using a convex-shaped function (approximately a cubic function), an approach similar to the one used in the speed-intensity calculation, with the key concept being that an impact of 4g is more than twice as hard on the body as an impact of 2 g. The weighted impacts were totaled and finally scaled to give more workable values expressed in arbitrary units (AU). | Arbitrary units (A.U.) | StatSports | Not provided |
| Impacts | Using the magnitude of the 3-dimensional accelerometer values at any time point, impacts were identified as maximum accelerometer magnitude values above Xg in a 0.1-second period in relation to manufacturers' specifications. <br><br> *GPS Sports*: 6 ranges of impacts according to the impact intensity: very light (<5.0-6g), light to moderate (6.1–6.5), moderate to heavy (6.5–7.0), heavy impact (7.1–8.0), very heavy (8.1–10.0) and severe (over than 10.1g). <br><br> *StatSports*: Values above 2g. <br><br> *RealTrack Systems*: Configurable threshold from 1 to 1000 G. | Count (n) | GPS Sports <br> StatSports | Not applicable |
| IMA | Application of polynomial smoothing curves between the start and end point of identified accelerative events. The magnitudes of such events are subsequently calculated by summing the accelerations under the polynomial curves, measured in terms of delta-velocity. | Meters per second square (m·s$^2$) | Catapult | Not provided |
| Impulse Load | Sum of the forces in the medio-lateral, anterior-posterior and vertical plane in relation to gravity. | Newtons (N) | Zephyr Technology | $\sum_{s=1}^{n} \frac{\sqrt{x_s^2 + y_s^2 + z_s^2}}{9.8067}$ |
| Locomotion Efficiency | To assess the within-match patterns of PlayerLoad$^{TM}$ and its individual planes in comparison to the locomotor activities, PLz was made relative to the total distance covered (TDC) as a measure of players locomotor efficiency. | Arbitrary units (A.U.) | Catapult | $\frac{\sqrt{\frac{(up_{t=i+1} - up_{t=i})^2}{100}}}{Total\ Distance\ Covered}$ |
| PlayerLoad$^{TM}$ (PL$^{TM}$) | Change in acceleration in the anterior-posterior (ax) medio-lateral (ay) and vertical (az) planes. | Arbitrary units (A.U.) | Catapult | $\sqrt{\frac{(fwd_{t=i+1} - fwd_{t=i})^2 + (side_{t=i+1} - side_{t=i})^2 + (up_{t=i+1} - up_{t=i})^2}{100}}$ |
| PlayerLoad$^{TM}$ x-axis (PLx) | Change in acceleration in the anterior-posterior (ax) plane. | Arbitrary units (A.U.) | Catapult | $\sqrt{\frac{(side_{t=i+1} - side_{t=i})^2}{100}}$ |

*(Continued)*

**Table 1.** (Continued)

| Index | Description | Units | Developing company | Formula |
|---|---|---|---|---|
| PlayerLoad[TM] y-axis (PLy) | Change in acceleration in the medio-lateral (ay) plane. | Arbitrary units (A.U.) | Catapult | $\sqrt{\frac{(fwd_{t=i+1}-fwd_{t=i})^2}{100}}$ |
| PlayerLoad[TM] z-axis (PLz) | Change in acceleration in the vertical (az) plane. | Arbitrary units (A.U.) | Catapult | $\sqrt{\frac{(up_{t=i+1}-up_{t=i})^2}{100}}$ |
| PlayerLoad[TM] 2D (PL2D) | Change in acceleration in the anterio-posterior (ax) and medio-lateral (ay) plane. | Arbitrary units (A.U.) | Catapult | $\sqrt{\frac{(fwd_{t=i+1}-fwd_{t=i})^2+(side_{t=i+1}-side_{t=i})^2}{100}}$ |
| PlayerLoad[TM] slow (PLslow) | Change in acceleration in the anterior-posterior (ax) medio-lateral (ay) and vertical (az) planes lower than 2G. | Arbitrary units (A.U.) | Catapult | $\sqrt{\frac{(fwd_{t=i+1}-fwd_{t=i})^2+(side_{t=i+1}-side_{t=i})^2+(up_{t=i+1}-up_{t=i})^2}{100}}$ |
| Player Load$_{RT}$ (PL$_{RT}$) | Vector sum of the Body accelerometric channel calculated through the sensorial fusion of inertial device sensors (accelerometer, gyroscope, magnetometer) in its 3 axes (vertical, anteroposterior and lateral). | Arbitrary units (A.U.) | RealTrack Systems | $PL_n = \sqrt{\frac{(X_n-X_{n-1})^2+(Y_n-Y_{n-1})^2+(Z_n-Z_{n-1})^2}{100}}$<br><br>$PL\ acummulated = \sum_{n=0}^{m} PL_n \ x\ 0,01$ |
| Player Load$_{RE}$ (PL$_{RE}$) | The player load is calculated and presented as a downscaled (i.e., divided by 800) value of the square sum of the high-passed filtered accelerometer values for the respective axes (X, Y, and Z). | Arbitrary units (A.U.) | ZXY SporTracking | $\frac{(x^2+y^2+z^2)}{800}$ |
| Total Load$_i$ | Total of the forces on the player over the entire session based on accelerometer data alone where aca is acceleration along the anterior-posterior axis, acl is acceleration along the lateral axis and acv is acceleration along the vertical axis, i is the current time and t is time. This is then scaled by 1000. | Arbitrary units (A.U.) | StatSports | $\sqrt{\frac{(aca_{t=i+1}-aca_{t=i})^2+(acl_{t=i+1}-acl_{t=i})^2+(acv_{t=i+1}-acv_{t=i})^2}{1000}}$ |

optimal players' physical fitness and avoid the occurrence of injuries due to an irregular workload dynamic between both sports contexts [7, 28]. Therefore, the sports tasks selection concerning the purpose of the training sessions and workload planning during competitive microcycles is fundamental for sports performance [39, 148].

In this review, a total number of 28 publications were found that performed an external workload analysis through accelerometry-based indexes in training and competition. Most studies analyzed the overall weekly workload (training and competition) and did not provide distinct training and competition hours so that the normalization is not possible making their comparison difficult [32, 62, 63, 89–92]. To solve this problem, different researches contain the workload related to playing time [39, 40, 64–66]. Therefore, future researches should provide training and competition hours or present the workload indexes both total and relative to playing time to allow for comparison between sports contexts.

Four studies that compared both sport contexts found a higher workload in training than in competition [25, 40, 64, 65]; four other articles reported the opposite [39, 41, 66, 93]. Higher competition workload reported in some studies may be the consequence of differences in weekly schedules, not accounting for conditions (e.g. day after game (starting vs substitutes), strength and power capabilities, technical-tactical elements, activation drills) for further analysis. Gentles et al. [41] analyzed the average training session workload in comparison with competition in university-level female soccer players through Impulse Load (20120±8609 vs. 12410 ±4067). Montgomery et al. [66] assessed the differences between 5vs5 game-based tasks in training in comparison with competition through PL/min (2.79±0.58 vs 1.71±0.84). Ritchie et al. [93] found a greater workload in training compared with matches during the pre-season (PL: 1985±745 vs. 1010±290), and the opposite during the competitive season (PL: 1014±383 vs. 1320±195).

On the other hand, if each training session is analyzed specifically and is not masked by overall weekly workload demands, a higher workload than the competition can be found

depending on the training session purpose. In this sense, Beenham et al. [40] found higher demands in 2vs2, 3vs3 and 4vs4 small-sided games in comparison with official matches in youth soccer measured by PL/min. Chandler et al. [65] and Fox et al. [64] showed a greater workload when the purpose of the training sessions was physical conditioning or game-based training in comparison with competition in women's netball and men's basketball respectively.

It is necessary for the best preparation of the player to understand physical and physiological stress during both training and competition [28, 66, 93]. A correct training session design related to the technical-tactical-physical purpose and the competition is important for workload managing during competitive microcycles [4, 5, 64]. In this sense, the use of effective strategies can help to anticipate the higher peak of performance in competition [67, 68, 94]. Also, it is important to monitor the player during all the training phases to assure the efficacy of the training effectiveness [95–97]. For this purpose, combined monitoring of internal responses with external workload demands through different variables based on tracking systems or accelerometry allow workload monitoring in an objective way [4, 15], being fundamental the selection of suitable workload indexes crucial for their control and also a clear presentation of the results for better decision-making by the team staff [149].

## Device location

In most of the analyzed studies, the inertial devices composed of tri-axial accelerometers for external workload monitoring in invasion team sports have been placed on the scapulae using an anatomically adjusted harness [8, 17, 26, 69, 93, 98, 99], except in a few studies where the companies recommend the location on the center of mass [24–26, 29] or the sternum [41].

The device location in team sports has been on the scapulae as this place is the most acceptable for detecting position coordinates by GNSS (latitude/longitude) in outdoor conditions [15, 40], or horizontal LPM using radio-frequency systems in indoor conditions [150–153]. Placing the device in a different location from the scapulae is sometimes selected because a body-worn accelerometer only measures the acceleration of the segment to which it is attached [154]. Thus, to detect more accurately the specific skills and workload of each sports discipline, accelerometers have been placed on different locations like the wrist in tennis [155], the head in swimming [156], the cockpit in kayak [157], the handlebar, seat or bike shank in cycling XCO-MTB [158, 159], or the tibia during running [19, 154, 160].

Therefore, if the aim is to record and measure specific events or skills, the device location should be the closest to the segment that performs the movement/action to achieve the highest accuracy [154]. Conversely, if the aim is to record and measure player tracking, the device must be placed on the scapulae to achieve the highest accuracy both in indoor and outdoor conditions [161–164]. To combine both measures and achieve the highest accuracy in both aims, the proposed solution is the development of a system composed of two interconnected parts: (a) an inertial device or HUB (signal concentrator) located on the scapulae for tracking location and receiving the signal from (b) different micro-sensors (accelerometers, gyroscopes, etc.) placed on different anatomical locations to detect the specific accelerometry-based workload of each segment. These micro-sensors would send the signal to the HUB by wireless technology (Ant+, Bluetooth, among others) where it would be stored for subsequent analysis. In this way, a recent study proposed the monitoring of different body locations simultaneously (scapulae, lumbar region, knees and ankles) through multiple inertial devices that could be attached to the body with elastic bands and harness or with a specific one-piece sport jumpsuit with pockets [165].

## Accelerometry-based workload indexes

Currently, from the raw data obtained by the accelerometer, the analysis of external workload is carried out from two main variables: impacts as a function of intensity ranges and $PL^{TM}$ in its different variants (2D, x-axis, y-axis, z-axis, slow). Workload quantification about the intensity of impacts has been used predominantly in rugby [92, 100–103], soccer [42] and American football [99]. In rugby and American football the detection threshold is 10G [91, 92, 99, 103]. In soccer, the detection threshold is 5G, and the number of impacts ranges from 490±309.5 to 613.1±329.4 number of impacts [42]. In trying score games, a greater number of >5G impacts were found in rugby 895±325 [102], rugby union 1222±607 [103] and American football 951 ±192 [99]. This difference could be due to the lower intensity of physical contacts in soccer (disputes, tackles, charges) compared with rugby or American football (collisions, scrum, rucks, etc.), or different game duration and format. Therefore, it is important to analyze the specific demands of each sport modality and to adapt the indexes or threshold detection for an accuracy workload monitoring during training and competition contexts.

The most frequently used accelerometry-based index is $PL^{TM}$, which represents the accumulative workload in the three axes of movement during all sessions. For the comparison between sports disciplines, the variable $PL^{TM}$ is normalized by total session time [min]. Studies that included time normalized $PL^{TM}$ reported different workloads across sports as soccer 10.18±2.12 [40] or 10.6–13.2±1.5–2.5 [43], netball 9.4–10.6±2.4–3.6 [70], MMA 15.37±1.71 [166], handball 9.18–9.76±0.6–1.4 [69], rugby union 7.6±0.6 [104], hockey 13.8–12.5±1.6–1.0 [44], lacrosse 7.6–9.9±1.5–2.7 [71], Australian football 15.1–16.3±0.9–1.4 [105] and rugby 7.2–10.4±0.8–2.0 [100]. These data confirm that each sport has specific demands regarding external workload, being the ranges as a consequence of the different playing positions. Therefore, it is necessary to analyze the competition workload to design the optimal stimulus during training sessions for sports performance enhancement.

Different investigations also use other $PL^{TM}$-dependent variables such as their segmentation by axes ($PL_x$, $PL_y$, $PL_z$) to analyze the specific contribution of each axis in the total workload on the technical-tactical skills or which axis is more related to fatigue during competition [40, 62, 65, 70, 106, 167]. Otherwise, $PL_{slow}$ quantifies the contribution of low-intensity workload (<2G) to the total workload of the players [90, 100, 104, 107, 108]. These two indexes allow higher accuracy and individualization of the demands performed by the athletes. The highest contribution to the external workload suffered by the athletes is from the vertical axis of movement, being over 50% of the total workload (y-axis > x-axis > z-axis). Also, the low-intensity workload represented between 35 and 50% of the cumulative PL. Therefore, the assessment of both indexes will be important for designing individualized technical-tactical-physical workloads and making possible the objective detection of players' deficiencies and optimum performance value enhancement.

Finally, concerning the company that develops each device, other variables are found in sport sciences area such as Dynamic Stress Load [45], Body Load [109], Total Load [46], Force Load [32], Impulse Load [41], $PL_{RT}$ [27] or $PL_{RE}$ [26] to quantify the cumulative workload during training sessions or official matches in team sports. These indexes are based on the accelerometry raw data in the 3-axes of movement applying different algorithms and scaled values. This makes the comparability of data from different devices difficult [168]. The result is a very high to perfect correlation between accelerometry-based workload indexes with very large differences in absolute values [47].

Thanks to this great number of variables, it is possible to specifically analyze the accelerometry load in each sports discipline, both the accumulative load and the specific demands of skills/abilities, with the aim of individualizing the specific load in each sport in relation with

player position or roles in competition. However, a consensus is necessary to be able to compare data among devices.

## Accelerometer technical features

Most studies with the purpose to detect movement patterns in invasion team sports through accelerometers presented a sampling frequency of 100 Hz. This technical feature is important to ensure high data quality during data collection [169]. A lower sampling rate is related to lower accuracy [153]. For this reason, Migueles et al. [170] recommended the use of a minimum of 90 Hz when researchers are using the manufacturer methods, or 100 Hz when researchers are filtering and processing the signal on their own. Therefore, a sampling frequency of 100 Hz is enough to detect external workload in the three-axis of movement through accelerometers in team sports.

Other important technical features that should be considered are the planes of movement (2 planes x-y vs 3 planes x-y-z), the number of accelerometers that compose the device and the output range of each accelerometer. Most of the studies shown that triaxial accelerometers composed the inertial device used. This characteristic is fundamental to detect three-dimensional movement and, consequently, to calculate the external workload index, which requires the acceleration in the three axes [15, 47, 170]. On the other hand, only 9 studies specified the number of accelerometers used in the devices. The number of accelerometers is only important if the output range of each accelerometer is considered. WIMU PRO is composed of four accelerometers with specific output ranges ±16g, ±16g, ±32g and ± 400g [47, 48, 72] while Optimeye S5 is composed of three ±16g accelerometers [49]. This technical feature is very important due to the second device cannot detect the peak of force generated when a collision is over than 16g. Therefore, the number of accelerometers cannot be considered as a quality criterion without the output range of the accelerometers that compose the device. For this reason, both technical aspects should be described in the methods section to identify if the accelerometers can detect with high accuracy all movements or events evaluated (total workload and peak workload) during training and competition.

Finally, the most important technical feature is the validity and reliability of accelerometers. The reliability is the consistency of measure between devices and across time that allows the workload comparison between devices and between sessions, while the validity is the extent to which the scores actually represent the variable they are intended to [171]. In this systematic review, it is worrying that only 25.4% of the selected studies reported both validity and reliability, 31.4% only reliability and 8.5% only validity of accelerometers. Specifically, the validity and reliability of PL and MinimaxX [16, 19], $PL_{RT}$ and WIMU PRO [18, 21], Body Load and SPI--PRO [172] and Impulse Load and Zephyr Bioharness [173] have been evaluated previously. All devices and accelerometry-based variables presented satisfactory results, except BodyLoad [172].

Among studies that cited the reliability and validity of accelerometers, 15 investigations (i.e. 12.7%) cited the reliability and validity of other devices that were not used in their respective research. Investigations measured with Optimeye and Team 2.5 devices (Catapult Sports) [62, 67, 69, 73, 74, 104, 110–112], ZXY Sportracking (Radionor Communications) [26], X8-mini (Gulfcoast Data Concept) [24, 25, 29], Actilife v12 (ActiGraph) [75], and Viper V2 (StatSport) [76] cited the validity study of Barrett et al. [19] and reliability of Boyd et al. [16] realized with MinimaxX devices (Catapult Sports). Noteworthy, 34.7% of the studies did not report the validity or reliability and did also not refer to literature findings for this purpose.

Therefore, the validity and reliability of the accelerometer-derived outcomes to determine how they can be effectively applied to individual and team sports is necessary. A consensus in

this aspect should be reached for that companies need to assess their devices through an independent and standardized protocol that assure the accuracy and reproducibility of accelerometer-derived outcomes in different context and sports.

## Sports modalities, sex and category

Most of the selected studies have been on Australian football, rugby, soccer and basketball. The rest of invasion team sports have aroused low research interest. Thanks to the Australian Centre of Microtechnological Research through the Project 2.5 "Technology of Communication to Athletes Monitoring" beginning to design a unique and non-intrusive device for sports monitoring in real-time in 2001 [14], the research topic has been centered on the most popular sports in this region (Australian football and rugby), developing specific variables such as impact/collision detection [92, 100–103]. Later, from the results obtained and the high socioeconomic impact, this technology began to be used in the most popular sports in Europe and the United States such as soccer and basketball [6, 27, 29, 40, 50, 51, 63, 64].

This socioeconomic aspect is also found in the sports category and sex. The majority of studies were performed at the elite and professional-level (77%) with men players (87%). This has meant that numerous studies have analyzed the relationship between accelerometry-based workload indexes and low-cost objective and subjective monitoring methods due to the low economic resources in the rest of the categories. Different research has related the accelerometry-based indexes with heart rate workload indexes such as training impulse (TRIMPS), and Edwards or summated heart rate zones (SHRZ) finding very high to almost perfect validity values [8, 17, 64]. Besides, it also has been related to sRPE [17, 52, 63] or subjective tools such as Integral System of Training Task Analysis (SIATE) [53, 174] with high to very high correlation values. Therefore, due to the low economic resources in non-professional categories and women's sports, these alternative methods could be used for workload monitoring subjectively, both at internal and external workload levels. In addition to finding alternative methods for workload monitoring, it is the task of researchers and professional teams to help knowledge development through research in these sports populations where the largest number of athletes and licenses are to be found.

Although there are existing correlations between accelerometry-based workload with external subjective (SIATE) and internal subjective (sRPE) and objective (TRIMPS, Edwards and SHRZ) workload indexes, the use of accelerometers is recommendable to quantify external workloads objectively. Their reliability, precision and sensitivity are greater compared to other external workload quantification systems such as automatic and semiautomatic time-motion analysis (video tracking, GNSS or LPM) [26, 30]. Automatic and semiautomatic TMA systems may underestimate the external workload demands because static high-intensity actions (jumps, collisions, etc.) are classified in the low-intensity actions group [26]. Therefore, recent investigations identified that microtechnologies (e.g. wearable microsensors and accelerometers) may represent a valid and practical alternative to TMA and offer distinct advantages compared with TMA such as the relative simplification to analyze data using either proprietary or used-defined algorithms that quantify movement, detect forces generated by the athlete related to gravity, the non-invasiveness, the measuring of internal and external workload simultaneously and the real-time feedback to minimize fatigue and injury risk while ultimately improving performance [18, 30, 33, 47, 149].

## Limitations

While the results of this systematic review have provided a global overview of accelerometry-based workload demands in invasion team sports, considering multiple factors such as

journals, context, categories, sex, body locations, brands and devices, technical features of accelerometers, variables and specific sports, some limitations to the study must be acknowledged. Firstly, only studies from Web of Science databases, PubMed and Scopus wrote in English were included, thereby potentially overlooking other relevant publications in other languages. Besides, although the study topic was invasion team sports, it would be interesting to include in a future systematic review all team and individual sports to achieve a better overview.

## Conclusions and practical applications

This systematic review shows all studies that carried out workload monitoring through accelerometry-based indexes in invasion team sports during training and competition contexts. From the findings of the present systematic review, different conclusions could be shown:

1. There has been an increase in workload monitoring through accelerometry-based indexes in training and competition, for which previous validity and reliability analysis is necessary both to evaluate the accuracy and allow comparison among and within units.

2. A large number of accelerometry-based workload indexes were found depending on the device manufacturing companies. The most widely used is $PL^{TM}$, but index unification among companies is required to be able to compare results among studies.

3. The upper back (scapulae) is the most common body location used to place the inertial device on the players due to the better tracking signal reception by Global Navigation Satellite Systems in outdoor and Local Position Measurement in indoor conditions. New research should quantify the workload not only on the scapulae but in different body segments simultaneously in training and competition contexts in order to identify the real workload of the athlete during skill performance and sport locomotion more accurately.

## Supporting information

**S1 Table. Selected articles in goal striking games.**
(DOCX)

**S2 Table. Selected articles in goal throwing games.**
(DOCX)

**S3 Table. Selected articles in try-scoring games.**
(DOCX)

**S4 Table. Quality criteria used to analyze the quantitative publications (extracted from Law et al. [37]).**
(DOCX)

**S1 File. PRISMA 2009 checklist.**
(DOC)

**S2 File. Search database output.**
(XLSX)

## Author Contributions

**Conceptualization:** Carlos D. Gómez-Carmona, Alejandro Bastida-Castillo, Sergio J. Ibáñez, José Pino-Ortega.

**Data curation:** Carlos D. Gómez-Carmona, Alejandro Bastida-Castillo.

**Formal analysis:** Carlos D. Gómez-Carmona, Alejandro Bastida-Castillo.

**Funding acquisition:** Sergio J. Ibáñez.

**Investigation:** Carlos D. Gómez-Carmona, Alejandro Bastida-Castillo, José Pino-Ortega.

**Methodology:** Carlos D. Gómez-Carmona, Alejandro Bastida-Castillo, Sergio J. Ibáñez, José Pino-Ortega.

**Project administration:** José Pino-Ortega.

**Resources:** Carlos D. Gómez-Carmona, Alejandro Bastida-Castillo, Sergio J. Ibáñez.

**Software:** Carlos D. Gómez-Carmona, Alejandro Bastida-Castillo.

**Supervision:** Sergio J. Ibáñez, José Pino-Ortega.

**Writing – original draft:** Carlos D. Gómez-Carmona, Alejandro Bastida-Castillo.

**Writing – review & editing:** Carlos D. Gómez-Carmona, Alejandro Bastida-Castillo, Sergio J. Ibáñez, José Pino-Ortega.

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
