## [Decision Letter · Decision Letter 0]

1 Apr 2020

PONE-D-19-33961

Accelerometry as a method for external workload monitoring in invasion team sports. A systematic review

PLOS ONE

Dear Mr. Gómez-Carmona,

Thank you for submitting your manuscript to PLOS ONE. After careful consideration, we feel that it has merit but does not fully meet PLOS ONE’s publication criteria as it currently stands. Therefore, we invite you to submit a revised version of the manuscript that addresses the points raised during the review process.

We would appreciate receiving your revised manuscript by May 16 2020 11:59PM. To enhance the reproducibility of your results, we recommend that if applicable you deposit your laboratory protocols in protocols.io, where a protocol can be assigned its own identifier (DOI) such that it can be cited independently in the future. For instructions see: http://journals.plos.org/plosone/s/submission-guidelines#loc-laboratory-protocols

We look forward to receiving your revised manuscript.

Kind regards,

Cristina Cortis, Ph.D.

Academic Editor

PLOS ONE

2. PLOS ONE does not permit references to unpublished data; therefore, we request that you either include the referenced data or remove the instances of "data not shown," "unpublished results," or similar.

4. Please include your tables as part of your main manuscript and remove the individual files. Please note that supplementary tables (should remain/ be uploaded) as separate "supporting information" files

Reviewers' comments:

Reviewer's Responses to Questions

**Comments to the Author**

1. Is the manuscript technically sound, and do the data support the conclusions?

Reviewer #1: Yes

Reviewer #2: Partly

2. Has the statistical analysis been performed appropriately and rigorously? 

Reviewer #1: N/A

Reviewer #2: N/A

3. Have the authors made all data underlying the findings in their manuscript fully available?

Reviewer #1: Yes

Reviewer #2: Yes

4. Is the manuscript presented in an intelligible fashion and written in standard English?

Reviewer #1: Yes

Reviewer #2: Yes

5. Review Comments to the Author

Reviewer #1: This systematic review provides insights regarding the use of accelerometers to quantify external workloads in team sports. The literature review is well-written and correctly formatted for submission to PLOS ONE; it is well organised providing clear and comprehensive information on the topic. A thorough analysis of current literature is provided identifying and discussing gaps on this topic; however, introduction section should be really improved to further underline the need to develop this systematic review. The current version of the manuscript clearly presents the literature background, without really pointing out why this review is needed. Furthermore, authors should develop the discussion section to point out the relevant information retrieved for practitioners. Please make sure to really edit the manuscript according to these overall suggestions. Furthermore, a better use of abbreviations is required throughout the manuscript. Specific comments are listed below.

Line 11 – “associate professor” should not be included here.

L41 – replace “colon” with “comma”.

L50 – Be consistent with the use of “load” or “workload” throughout the manuscript. Edit accordingly within the entire manuscript.

L60 – Please replace “rated” with “rating of”.

L61 – Remove “on the other hand”. This is too colloquial.

L62 – Include “various variables like” after “through”.

L63-64 – Please try to link the first and second paragraph of the introduction section. It appears that you completely

L70 – Specify what are these instruments considered as “gold standard” and “criterion measures”.

L82 – Please specify that you are referring to “internal” load.

L84-85 – This sentence is not clear. Please edit for clarity.

L89 – Include the following reference here.

- A Review of Player Monitoring Approaches in Basketball: Current Trends and Future Directions. Fox JL, Scanlan AT, Stanton R. J Strength Cond Res. 2017 Jul;31(7):2021-2029. doi: 10.1519/JSC.0000000000001964.

L93 – “rated perceived exertion” should be “RPE”. Be consistent when using abbreviation within the manuscript and check the other abbreviations used. Make sure to explain it the first time you use it and then use the abbreviation.

L94-96 – A reference is required here.

L96-99 – This is a very important point to be discussed here. Please consider also to include something about the comparison between accelerometers and time-motion analysis. With this regard, accelerometers do not provide information about static actions too when an effort is performed without an acceleration (e.g. screenings or a prolonged stance position in basketball) but their reliability is greater compared to other systems. Furthermore, accelerometers provide real-time data. This paragraph should be improved to better develop the importance of using accelerometers compared to other systems and to further underline the importance of carrying out this review. For example, it has been stated that “microtechnologies (e.g., wearable microsensors and accelerometers) in basketball may represent a valid and practical alternative to TMA” (Ferioli, 2020) or “accelerometers offer distinct advantages compared with TMA, with accelerometer data being relatively simple to analyze using either proprietary or user-defined algorithms that quantify movement” (Fox, 2017).

- Match Activities in Basketball Games: Comparison Between Different Competitive Levels. Ferioli D, Schelling X, Bosio A, La Torre A, Rucco D, Rampinini E. J Strength Cond Res. 2020 Jan;34(1):172-182. doi: 10.1519/JSC.0000000000003039.

- A Review of Player Monitoring Approaches in Basketball: Current Trends and Future Directions. Fox JL, Scanlan AT, Stanton R. J Strength Cond Res. 2017 Jul;31(7):2021-2029. doi: 10.1519/JSC.0000000000001964.

L115 - Should “retrieved” be “peer-reviewed” manuscript?

L128 - Why did the authors include only two electronic databases? The inclusion of other databases (like Scopus, MEDLINE, ERIC, Google Scholar, SCIndex, ScienceDirect..) would have been more appropriate.

L130 – In line with my previous comment, including more keywords (e.g. microtecnologies, workloads, physical demands..) would have been more appropriate.

L140 – Replace (iii) with “(3)”

L141 – Replace (3) with “(4)”

Figure 2b – “Int J Sport Physiol” should be “Int J Sports Physiol Perform”; “J Sport Sci” should be “J Sports Sci”. Please check all the journal abbreviations and edit if needed.

L261-263 – The abbreviations included here have not been explained before. Consider including the variables without abbreviation and/or include something like “see Table 1 for definitions”.

L273 – “PlayerloadTM” should be PLTM. (see line 244). The authors should check the use of correct abbreviations (when required) throughout the manuscript.

L307-312 – I would suggest including a message for practitioners regarding the use of accelerometers when monitoring workloads during training or competitions. The current version of this paragraph does not provide any relevant information, while this systematic review should provide insights on this topic. For example, is it important to monitor external workloads during both competitions and trainings? Would the monitoring of competition workloads assist planning training workloads? The authors did a big effort investigating workload sustained during both competitions and trainings, as such they should provide some relevant information here to underline the utility of this systematic review.

L320 – definitions of GPS and GNSS are required here.

L323 – definitions of LPM and UWB are required here.

L337 – definition of HUB is required here.

L331-342. This paragraph is well discussed, providing relevant information and practical applications. Well done.

L340 and 344 – Neuromuscular load – Here the authors are including the term “neuromuscular load” that was not included in the introduction or elsewhere before. Can this term be replaced with external workload as done in the introduction? If not, the introduction should be edit accordingly.

L350-351 – Are the authors referring to soccer competitions? This is not clear.

L359 – Again, “PlayerloadTM” should be PLTM. (see line 244).

L381-382 – “making possible objective players’ weaknesses detection” is not clear enough. Please edit for clarity.

L390-392 – This is a systematic review, as such unpublished data should not be included. Remove this sentence. It might be more appropriate to include something regarding the development of a strategy to correctly use and interpret data retrieved with different systems (and algorithms).

L416 – Include the following reference here:

- The relationships between internal and external training load models during basketball training. Scanlan AT, Wen N, Tucker PS, Dalbo VJ. J Strength Cond Res. 2014 Sep;28(9):2397-405. doi: 10.1519/JSC.0000000000000458.

L417 – RPE was already abbreviated.

413-423 – The authors should consider that despite the existing associations, accelerometers are used to quantify the external workloads, while the cited tools are used to measure the internal workloads. This should be underlined, as the reader could erroneously interpret that the systems are interchangeable. This is not the case, as workload quantification should include both internal and external measures. It might be of interest to include here and discuss the methodologies used to quantify external workloads (TMA, tracking systems..) highlighting the positive and negative aspects of using accelerometers than other systems. This paragraph should be really edit accordingly.

Reviewer #2: Thank you for giving the chance to review the manuscript on accelerometry as monitoring tool in team sports.

General comments:

The manuscript seems to fulfill the journal publication criteria and addresses a topic perceived relevant by the scientific community for training monitoring and regulation. To a large extent, the manuscript documents descriptive data of the selected studies. The interpretation and conclusions for practical relevance can be improved. Otherwise, the value of this review is questionable. A good example of a clearly outlined relevance of this manuscript’s contribution to the state of knowledge and community is device location. Here, the manuscript clearly can recommend a most common and best practice. In other regards, the value added by this review (except for the documentation of descriptive numbers) seems often questionable. In general, the value of the reported numbers can be improved by deepening the interpretation and conclusions drawn from these numbers.

A second general concern is the way how the relevance of this review and objectives is displayed in the introduction. It is recommended to outline the current deficits and needs that justify this review rather than building strongly on an increased interest (see L100), which is by the way one of the results and not a given fact at the stage of the introduction (at least it is not supported by the literature). This makes it a weak justification.

Other general concerns are addressed in specific comments relating to concrete examples but should be considered throughout the manuscript. The whole manuscript requires to be checked for weaknesses comparable with the examples pointed out in the specific comments.

The manuscript can be suitable for further consideration after a major revision. Perhaps, an expert in scientific English writing should be consulted.

Specific comments:

ABSTRACT

L30-31: „for using it if they are coherently organised” is no valuable information. More valuable would be to add the benefit of applying the method. For example: “A rapidly increasing number of studies supports the practical implementation of accelerometry monitoring to regulate and optimise training schemes.” (If such statement is in line with the literature and the authors intentions).

L33: “and identify and organise them”: First, correct is "TO identify and organise them". Second, these are weak objectives. Consider something like “to reflect the current state of knowledge and to conclude recommendations for application and scientific investigations".

L38-39: The subsequent list is not a list of “fulfilled aspects”. Reword (e.g. “main findings include”). The list is also unclear; it is not self-explaining that the first item is the addressed issue (e.g. body location for device), followed by the specification/most frequent finding etc. (e.g. scapulae). Also the punctuation marks are inconsistent (i.e. “,” vs “:”).

L42-44: Please aim for condensed language, straight to the point: E.g. “A great number of variables and devices from various companies makes comparability between findings difficult; unification is required.” Also be careful with terms like “evident”.

INTRODUCTION

L52-54: First, no “,” before “that”. Second, the information “that are analysed by sport scientists and fitness coaches” is not very relevant and can be removed. Third, “with the purpose of explaining" can be replaced by “to explain". Fourth, “explaining training effects”: Really? Maybe more appropriate would be “regulate training volumes” or something comparable. Fifth, “injury risk or that of overtraining”: Reword as “risk of injuries and overtraining”.

Please take this first sentence as an example how to improve the writing throughout the whole manuscript.

L55: “respond” is correct.

L64: Be more precise. It is not an area. Consider “At high levels”

L65: Shorten: “to obtain” instead of “with the aim of obtaining”

L66-67: The reviewer thinks that it is not performance that is difficult to be measured but developments, training activity, input-outcome etc. Consider rewriting this statement because, as it is, this is not the reason for the subsequent statement (i.e. different methods to measure workloads indirectly).

L77: Do not use “among others”. Consider “include A, B, C”. Then, there is no need for “among other”, “and more”, “etc.”.

L79: no “,” before “such”.

L100: the “increasing interest” is not a given fact at this stage and is not supported by the literature. It is a part of the result of this study. So, please do not refer to it as a reason for this study. Instead, you may reason by benefits of accelerometry over other methods as you indicated in the comparison with video tracking methods.

L101 ff: Please shorten the purpose and focus on things that are relevant to clarify what the major purpose is. Do not redundantly use 2 sentences introducing different purposes. The purpose and objective should match the need analysis for this study (i.e. be in line with its relevance). For example, the purpose is to summarize/reflect the current state, outline best practices, and conclude recommendations.

METHODS

L113-116: Another example for condensing the manuscript: Please consider to merge both sentences into “This manuscript is a systematic review that analyses [30] scientific articles related to workload monitoring via accelerometry in sports.” Note that you downgrade your own work by stressing it is an exclusively theoretical work.

L116-118: Please remove completely. Repeating systematic methodology is self-explaining in a systematic review and having no complex statistics is again downgrading your own work. There is no point in doing such statement.

L119-125: Such questions do not belong in this chapter; they should be clarified in the introduction. The reviewer thinks they are clear. Second, they are not required for justification of the inclusion criteria. Therefore, the reviewer suggests removing this part. This allows merging this sub-chapter with the subsequent one: One chapter "Design and Search Strategy". You may start with the sentence suggested in the previous comment and then go on with L128 introducing the database search.

L132: Replace “or” by “and”

L139: Consider “across ages” or replace “different” by “various”. Otherwise, “different” suggests that it was a criterion to have at least 2 levels in each study and these levels must differ. Maybe the best would be to simply remove it. As the reviewer understood it, the level was in fact not a criterion. Studies were included “irrespective of level” (that could be another wording you may use).

L140: Same goes for sexes (see previous comment). Both sexes separately and the combined analyses of both were included in the review. So, sex is no inclusion/exclusion criterion. Also, what is this different format of the number (i.e. “iii”)?

L150: Please adjust by saying “competitive sports context” because it would be wrong to say health activities and practices implemented in elderly cannot be sport! Please remove “older people” in brackets.

L152: Remove the sixth point. It is redundant after defining invasive team sports as inclusion criterion previously.

L158: Please change “competition (playing level and sport)” to “sport and competition level”. Moreover, check throughout the rest of the manuscript to use “competition level” instead of players’/playing/sports level.

L168-170: Please shorten to “by five university full professors with a PhD in sports science and a large number of publications in the field of xy [specify; 1-2 words].”

L176: Do you mean “rated” instead of “scored”?

RESULTS

L192 ff: You may shorten as follows: “From the remaining 296 articles, 83 did not fulfill the inclusion criteria and were removed after revision of the abstract and another 141 after full-text assessment.”

L201: Consider to shorten: “Main reasons for exclusion were non-competitive sports contexts (n=41) and reliability and validity analyses of raw data and load indexes (n=37)”.

L203-205: If these two reasons’ numbers are mentionable, please add the numbers (n=?). Otherwise, there is no need to mention these two reasons at all.

L210-217: The usage of Kappa, thresholds for quality levels, and 95% CI should be introduced in methods section.

L232: Remove “With respect to the sports context”. Just start with “Most studies […]”.

L237: Change to “males” and “females” instead of “men” and “women” as you mention that some analysed juniors.

L244: Most FREQUENTLY used.

L259-260: Again, remove “with respect…”. Change to “x, y, and z were the most frequently investigated invasion team sports.”

L260: What do you mean by “in a specific manner”?

L267: As mentioned in general comments: to “select” and describe is a weak purpose. Consider “summarise” and something like “identify common practices and conclude recommendations” based on the review. For explanation: A pure selection and reflection of descriptive numbers is not very helpful for the community except for other manuscripts to back up statements like “commonly/most frequently used”. A more valuable contribution is to draw conclusions and to outline recommendations based on these numbers.

L269: Replace “Thus” by “Therefore”. Please be aware that they are not interchangeable as thus strongly relates to the way how something is done.

L271: Be precise and correct in language: The evolution is not rapid. The increase may be. Here, the reviewer suggests “show a rapidly increasing number of publications”

L276: “Most studies” instead of “Most of the studies”

L277: “limited” instead of “reduced”

L279: “occurrence” instead of “appearance”

L283: Add: a total NUMBER of 16 publications

L284-286: Please clarify here if these studies provided training and competition hours so that they allowed for normlisation or not. If not, state it here.

L286: Rephrase: “Four studies that compared both sport contexts found higher workload in training than in competition; three articles reported the contrary.”

L290: Please clarify “in relation to their purpose”

L292: “female” instead of “women”

291+294+302+305: Probably you mean “in comparison with” instead of “with respect to”. Please be aware that these are not interchangeable.

L295-298: Please rephrase and condense the findings from Ritchie et al. Writing “they compared a with b, finding higher values in a and lower values in b” is not recommended.

L310: Consider “suitable” instead of “necessary”.

L323: If the reviewer is not mistaken, LPM stands for Local Position Measurement (you did not introduce this and other abbreviations) and is not defined by the system that is applied to identify the location. Please check and correct in case.

L334: The reviewer suggests being careful with “must” in this context. Consider “should” or “is recommended”

L344: Is it really NEUROMUSCULAR load?

L351: Sentence is unclear. Add units to impact numbers. What means “greater impacts ARE FROM x±y”? Rephrase. Reword “under-19”/”under-15”.

L353-355: Change to “due to lower intensity of physical contacts in soccer compared with American football”

L356: Therefore instead of thus

L359: most FREQUENTLY used. Also, replace “that” by “, which” (with comma)

L360-367: Rephrase these two sentences. For example: “For comparison between sports disciplines, the variable PLTM is normalized by total session time [min]. Studies that included time normalized PLTM reported different workloads across sports (soccer: x, netball: y…)”

L372: Reword “which of them most contributes”.

L375-376: Reword “plane where the movement is produced”.

L378-382: Shorten and clarify the sentence and statements made.

L386: Shorten to “These indexes”

L387: Change “but with the implementation of” to “applying”.

L388: Simplify (e.g. “This makes the comparability of data from different devices difficult. The result is a […]”.

L440-441: The review did not assess the state of validity and reliability of workload monitoring and cannot draw conclusions if the existent research sufficiently covers this concern, right? You may elaborate this in an appropriate section to allow such conclusion. The reviewer recommends such addition rather than removing this conclusion to strengthen the value of the current manuscript.

L447: “LPS” seems to be a typo. Probably, “LPM” was meant.

6. PLOS authors have the option to publish the peer review history of their article (what does this mean?). If published, this will include your full peer review and any attached files.

Reviewer #1: No

Reviewer #2: Yes: Philip X. Fuchs

---

## [Author Response · Author response to Decision Letter 0]

16 May 2020

Dear Editor,

We have carefully considered all reviewers' considerations of the paper (PONE-D-19-33961). Please find enclosed our detailed answers to reviewers' queries. The authors declare that the manuscript is original and has not been considered for publication elsewhere. Additionally, the authors had approved the paper for release and agree with its content.

All responses to the specific reviewers and editor comments are attached in the document Response_Reviewers.pdf

Kindly regards

---

## [Decision Letter · Decision Letter 1]

8 Jun 2020

PONE-D-19-33961R1

Accelerometry as a method for external workload monitoring in invasion team sports. A systematic review

PLOS ONE

Dear Dr. Gómez-Carmona,

Thank you for submitting your manuscript to PLOS ONE. After careful consideration, we feel that it has merit but does not fully meet PLOS ONE’s publication criteria as it currently stands. Therefore, we invite you to submit a revised version of the manuscript that addresses the points raised during the review process.

We look forward to receiving your revised manuscript.

Kind regards,

Cristina Cortis, Ph.D.

Academic Editor

PLOS ONE

Reviewers' comments:

Reviewer's Responses to Questions

**Comments to the Author**

1. If the authors have adequately addressed your comments raised in a previous round of review and you feel that this manuscript is now acceptable for publication, you may indicate that here to bypass the “Comments to the Author” section, enter your conflict of interest statement in the “Confidential to Editor” section, and submit your "Accept" recommendation.

Reviewer #1: (No Response)

Reviewer #2: All comments have been addressed

2. Is the manuscript technically sound, and do the data support the conclusions?

Reviewer #1: Yes

Reviewer #2: Yes

3. Has the statistical analysis been performed appropriately and rigorously? 

Reviewer #1: Yes

Reviewer #2: N/A

4. Have the authors made all data underlying the findings in their manuscript fully available?

Reviewer #1: Yes

Reviewer #2: Yes

5. Is the manuscript presented in an intelligible fashion and written in standard English?

Reviewer #1: Yes

Reviewer #2: Yes

6. Review Comments to the Author

Reviewer #1: The authors did a good job editing the manuscript in accordance with my previous comments and implementing the number of scientific databases revised for this systematic review The rationale of the study within the introduction section, the results section and some paragraphs of the discussion section have been better developed. In my opinion, some amendments are still required to improve the quality of the manuscript. In addition, I would suggest a further proofread of the entire manuscript as it still presents some grammatical errors and, in some cases (especially in the discussion section), poorly fluid and clear sentences.

Continuous line numbering has not been used, please remember to do this step in your future submissions as stated in the authors’ guidelines.

Specific comments are listed below.

ABSTRACT

Page 2, Line 4-6. Please include “1)” and “2)” before each aim.

P2, L17. “due to it is” does not sound very fluent. Please edit.

INTRODUCTION

P2- first line. “Load” should be “workload”. I still noticed the use of “load” in some sentences. Please replace it with “workload” where needed.

P2- second line. “decreasing” should be “to decrease”.

P2 – line 5-6 of introduction. “Workload” instead of “load”.

P3 L4-5. This sentence should be edited stating that current literature suggests to adopt strategies to monitor and quantify both internal and external workload sustained during training and competitions.

P3 L14. I do not agree with authors stating that current methodologies are “without restrictions”. This should be removed as this is not true. For example, in some professional basketball leagues the use of microsensors during competitions is not allowed (e.g. basketball).

P4 L8-12. This sentence is very hard to read and should be edited and/or splitted.

P4 L12-14. I do not fully agree with this statement as TMA in some cases can be more adequate for workload (and not “load” here - edit) quantification. Authors should consider differentiating manual than automatic and semi-automatic systems of TMA. Manual TMA can be used to identify static movements performed with high-intensity efforts (e.g. body contacts) better than accelerometers, while I agree that automatic video-based analysis can underestimate workload. Edit this section accordingly.

RESULTS

P7 L3. Should “1373” be “1371”? Please check articles numbers in this section.

DISCUSSION

P12 L2-4. Please do not repeat the aim of the study here. Include a general statement to resume what has been included in this systematic review.

P12 L25. “To possibility” should be “to permit”. Please edit.

P12 L26-29. The contrasting results may be also a consequence of different weekly schedules adopted. Competition formats (e.g. 1 game per week vs 2-3 games per week) considerably affect the workload outcomes, thus I suggest including this reasoning to justify the differences between these studies.

P14 L8-9. The abbreviation “GNSS” and “LPM” are required here. Please check the entire manuscript for the use of correct abbreviations.

P15 L10. Please edit with something like: the impacts registered >5G range from .. to ..

P15 L13. “in soccer” is repeated twice.

P15 L10-14. These differences can be also a consequence of different game duration and format.

P15 L21-25. If reporting ranges, include first the lower value and then the higher (i.e. 7.6-9.9 and not 9.9-7.6). Furthermore, “y” after ref 46 should be removed.

P15 L26. “Neuromuscular” should be replaced with “external”.

P15 L18-28. Can these ranges be a consequence of the different playing position? Consider including this reasoning here.

P15 last line. “and which is most..” appears to be quite disconnected. Please rephrase and include reference.

P16 L2-6. This sentence is too general; authors should include how to reach these results.

P16 L9. Accumulative or cumulative?

P16 L22-23. This sentence does not appear grammatically correct. Please edit for clarity.

P16 Last sentence. Please rephrase this sentence for clarity.

P17 L4-6. This sentence should be rephrased as it is not fluid and clear.

P17 L8-14. While I can understand the reasoning of authors here, the readers could be quite confuse from these sentences. Please rephrase for clarity.

P17 L17. Allows

P17 L19-21. This sentence is not clear enough. Are you referring that only 25.4% of studies reported BOTH validity and reliability?

P17 L21-25. Please rephrase this sentence as there are too many repetitions. Make this sentence clearer.

P18 L4-7. Maybe consider including “in different context and sports” to conclude this sentence.

P19 L9-11. I do not agree with this reasoning here. sRPE model is applied to quantify internal workload, which is a different variable than external workload. Why should we compare two different models measuring different variables? Please remove any comparison of accelerometers with sRPE model.

P19 L11. This comment refers to the entire manuscript. Consider a better use of RPE as in some occasions it has been used wrongly. Overall, RPE is not a load measurement; it is the rating of perceived exertion. Session RPE (sRPE) is the rating of perceived exertion for an entire training/competition session. sRPE workload is the workload quantified using the sRPE method. It is fundamental to use correct terms within a systematic review. See the following reference and edit within the manuscript accordingly.

Reference: Impellizzeri FM, Marcora SM, Coutts AJ. Internal and External Training Load: 15 Years On. Int J Sports Physiol Perform. 2019 Feb 1;14(2):270-273. doi: 10.1123/ijspp.2018-0935. Epub 2019 Jan 6.

P19 L11-21. According to my previous comment (P4 L12-14) I do not fully agree with this statement as TMA in some cases can be more adequate for workload (and not “load” here - edit) quantification. Authors should consider differentiating manual than automatic and semi-automatic systems of TMA. Manual TMA can be used to identify static movements performed with high-intensity efforts (e.g. body contacts, low jumps) better than accelerometers (that do not measure low static movements without important accelerations), while I agree that automatic video-based analysis can underestimate workload. Edit this section accordingly.

Reviewer #2: The reviewer thanks for the quick and solid revision of the submitted manuscript. The authors addressed all comments and did a great job in the implementation. Some of the new additions are very valuable.

There are few remaining concerns related to old and new parts of the manuscript that can easily be fixed. After such minor revision, the reviewer suggests that there is no further reviewing round required.

First, there are issues with the pdf file received for revision that make revision difficult. The reviewer cannot tell if this is due to technical problems or caused by the authors. Therefore, addressing both authors and editor, the problems should be fixed in future submissions and reviewing processes. The problems are:

1. The pdf for revision does not include line numbers any longer.

2. The changes highlighted in red to not show the original text that was removed (in contrast to the changes highlighted in blue; these changes were completely tracked).

3. Some of the previous reviewer’s comments are incorrectly presented. Examples are:

3.1. R2_SC5. L52-52: The reviewer’s comment was not “Third, “to explain” can be replaced by “to explain” as it was claimed. The reviewer’s comment was “Third, “with the purpose of explaining" can be replaced by “to explain".”

3.1. R2_SC8. L65: The reviewer’s comment was not “Shorten: “to obtain” instead of “to obtain””. The reviewer’s comment was “Shorten: “to obtain” instead of “with the aim of obtaining””

Due to missing line numbers, the reviewer apologies to refer to chapters, paragraphs, and sentences.

Second, to the authors: When responding to comments in such way “We really appreciate your suggestions. All of them have been considered to improve the final version of the manuscript”, please inform where and how such considerations were implemented, especially for general comments that do not refer to specific lines.

Abstract

Correct is as it was suggested “A great number […] makeS” because ONE number is the subject in this sentence.

Methods

Study design and Search strategy

1. Sentence: Change “peer-reviewed manuscripts (scientific papers)” to “peer-reviewed, scientific papers” because, technically, a manuscript is not the same as a paper and not all peer-reviewed products are scientific products and vice-versa; plus, it is shorter.

3. Paragraph: “genres” is assumed to be a typo. Change to “gender” or “sex” and be consistent in which one of the two is the appropriate to be used in this manuscript.

Quality of the studies

1. Paragraph: The last sentence “In the present study, a value of 0.93 was obtained […]” is a result of this study and should be moved to the results section. In the methods, only state that Kappa and 95% CI were calculated.

Results

Competition vs. Training

2. Paragraph: 2. Sentence: Change to “Most studies […] and DID not PROVIDE DISTINCT training […]”

2. Paragraph: Last sentence: Change “to possibility the comparisons between sports contexts” to “to allow for comparison between sports contexts”

3. Paragraph: 1. Sentence: The reviewer thinks it should be “in training than IN competition”. Also, remove “but” and write “four OTHER articles”.

3. Paragraph: 2. Sentence: It is clarified now, but the reviewer suggests rephrasing as follows: “[…] individualize the training sessions accounting for conditions (e.g. day […]”

3. Paragraph: Last sentence: Please make a full stop before and after Montgomery’s findings. Then replace Ritchie’s findings by: “Ritchie et al. [51] found greater workload in training compared with matches during the pre-season (PL: 19851985±745 vs. 1010±290) and the opposite during the in-season (PL: 1014±383 vs. 1320±195).”

Device location

2. Paragraph, new change: There is a wrong full stop after “[13,46]” and before “or by”; remove it. Change “Local Position Measurement (x/y axis)” to “horizontal Local Position Measurement (LPM)”.

Accelerometry-based workload indexes

1. Paragraph, middle: Change “are from” to “range from”. What are these numbers used in soccer? If it was the number of occurrences, then write “the NUMBER OF impacts”. If it was the actual impacts, use a unit.

2. Paragraph: There seems to be a mistake where, in the list of sports, one sport between soccer and netball is referred to as “y”. It is the one backed-up by reference number 84.

Accelerometer technical features

2. Paragraph, 2. Sentence: First, change to “Most studies”. Second, the sentence is grammatically wrong, and its meaning is unclear. Consider: “Most studies showed that triaxial accelerometers use inertia sensors”

2. Paragraph, 3. Sentence: Change to “[…] detect three-dimensional movement and, consequently, to calculate the external workload index, which requires the acceleration in the three axes”.

2. Paragraph, 4. Sentence: Change to “[…] studies specified the number of accelerometers used in the devices.”

2. Paragraph, 5. Sentence: Replace by “Moreover, the output range of each accelerometer is important and should be specified.”

2. Paragraph, 6. Sentence: it states “four accelerometers with different output ranges” but then two are identical (i.e. ±16g). Please correct by removing “different”.

3. Paragraph: That is indeed worrying. Please add one short clarification if studies, that did not report validity and reliability, did refer to literature reference for this purpose instead (yes/no?). In the subsequent paragraph you state that some studies did refer to inappropriate investigations. That means some do use references. So, simply state how many did (in percentage of the reviewed studies). That is enough.

7. PLOS authors have the option to publish the peer review history of their article (what does this mean?). If published, this will include your full peer review and any attached files.

Reviewer #1: No

Reviewer #2: Yes: Philip X. Fuchs

---

## [Author Response · Author response to Decision Letter 1]

13 Jun 2020

Dear Editor,

We have carefully considered all reviewers' considerations of the paper (PONE-D-19- 33961_R1). Please find enclosed our detailed answers to reviewers' queries. The authors declare that the manuscript is original and has not been considered for publication elsewhere. Additionally, the authors had approved the paper for release and agree with its content.

All answer to reviewers' queries was addressed in the Response Reviewers document.

Kindly regards

---

## [Decision Letter · Decision Letter 2]

2 Jul 2020

PONE-D-19-33961R2

Accelerometry as a method for external workload monitoring in invasion team sports. A systematic review

PLOS ONE

Dear Dr. Gómez-Carmona,

Thank you for submitting your manuscript to PLOS ONE. After careful consideration, we feel that it has merit but does not fully meet PLOS ONE’s publication criteria as it currently stands. Therefore, we invite you to submit a revised version of the manuscript that addresses the points raised during the review process.

Both reviewers suggested accepting the paper after few minor corrections. Therefore, I invite the authors to carefully consider all the suggestions and make a final checks for syntax and grammar.

We look forward to receiving your revised manuscript.

Kind regards,

Cristina Cortis, Ph.D.

Academic Editor

PLOS ONE

Reviewers' comments:

Reviewer's Responses to Questions

**Comments to the Author**

1. If the authors have adequately addressed your comments raised in a previous round of review and you feel that this manuscript is now acceptable for publication, you may indicate that here to bypass the “Comments to the Author” section, enter your conflict of interest statement in the “Confidential to Editor” section, and submit your "Accept" recommendation.

Reviewer #1: All comments have been addressed

Reviewer #2: All comments have been addressed

2. Is the manuscript technically sound, and do the data support the conclusions?

Reviewer #1: Yes

Reviewer #2: Yes

3. Has the statistical analysis been performed appropriately and rigorously? 

Reviewer #1: Yes

Reviewer #2: Yes

4. Have the authors made all data underlying the findings in their manuscript fully available?

Reviewer #1: Yes

Reviewer #2: Yes

5. Is the manuscript presented in an intelligible fashion and written in standard English?

Reviewer #1: Yes

Reviewer #2: Yes

6. Review Comments to the Author

Reviewer #1: The authors did a big effort editing the manuscript during the revision process. The manuscript has been greatly improved and will be a valuable addition to the literature. In my opinion, the manuscript should be considered acceptable for publication. I would recommend a final check for grammar, spelling, and punctuation mistakes during the editorial editing/formatting process (e.g. page 3, line 5 “suggestS”).

Reviewer #2: General comment:

The authors addressed all comments and fixed most issues. Some new mistakes emerged in the adaptations. For all cases, the reviewer provides concrete solutions.

In the specific comments, the reviewer refers to page and line numbers of the revised manuscript showing the tracked changes.

The reviewer suggests acceptance of the manuscript after the implementation of the provided solutions.

Specific comments:

Methods

P5, L2: Change “manuscript” to “paper” as the currently submitted piece of work is a manuscript (L1) that reviews published papers. This is how terms are correct. When published, a manuscript is a paper.

Discussion

P13, L25: A new error appeared after revision: Correct is “did not provide” (not: provideD)

P14, L3-4: The meaning of the new adaptation is unclear to me (studies are no consequence of different schedules!). The sentence is also grammatically incorrect. The reviewer can just assume that the authors meant something like: “Higher competition workload reported in some studies may be the consequence of differences in weekly schedules, not accounting for conditions […]” If this message is in line with the authors’ intentions, then I recommend to use this phrase.

P14, L15+16: 1. In both brackets you need to write “(PL: [numbers])”, otherwise no one knows what these numbers are. 2. You may change “in-season” to “competitive season”.

P15, L14+15: Why was “Global Navigation Satellite Systems” and “Local Position Measurement” replaced by short terms? Check if the short terms have been introduced previously, otherwise introduce them here

P15, L15: Remove “(x/y axis)” because, first, this is redundant and, second, horizontal is the better description anyway because x/y is a matter of definition and could be anything (therefore, would require a definition).

P16, L16-18: Previous suggestion was misunderstood, and adaptations are not correct anymore (the threshold does not range!). Change to full stop after the rugby and American football sentence (L16). Then, “In soccer, the detection threshold is 5G, and the number of impacts range from 490±310 to 613±329 [83].”

P18, L7: Change LESS sampling rate and accuracy to LOWER sampling rate and accuracy.

P18, L22-24: Remove “Moreover” and “that compose the device” and change to “The number of accelerometers is only important if the output range […]. Do not mess up the order in output range/range output.

P19, L1: Here is also incorrectly written “range output”. Change to “output range”.

P19, L19-23: Change to “Among studies that cited the reliability and validity of accelerometers, 15 investigations (i.e. 12.7%) cited the reliability and validity of other devices that were not used in the respective research.”

P19, L28-29: Change to “NOTEWORTHY, 34.7% of THE studies DID not report [not: reportED!] THE validity or reliability and did also not refer to literature FINDINGS for this purpose.”

7. PLOS authors have the option to publish the peer review history of their article (what does this mean?). If published, this will include your full peer review and any attached files.

Reviewer #1: No

Reviewer #2: **Yes: **Philip X. Fuchs

---

## [Author Response · Author response to Decision Letter 2]

3 Jul 2020

Dear Editor,

We have carefully considered all reviewers' considerations of the paper (PONE-D-19-33961_R2). Please find enclosed our detailed answers to reviewers' queries. The authors declare that the manuscript is original and has not been considered for publication elsewhere. Additionally, the authors had approved the paper for release and agree with its content.

All responses to reviewers' considerations have been attached in Response to Reviewers document.

Kindly regards

---

## [Decision Letter · Decision Letter 3]

13 Jul 2020

Accelerometry as a method for external workload monitoring in invasion team sports. A systematic review

PONE-D-19-33961R3

Dear Dr. Gómez-Carmona,

We’re pleased to inform you that your manuscript has been judged scientifically suitable for publication and will be formally accepted for publication once it meets all outstanding technical requirements.

Kind regards,

Cristina Cortis, Ph.D.

Academic Editor

PLOS ONE

Additional Editor Comments (optional):

Reviewers' comments:

Reviewer's Responses to Questions

**Comments to the Author**

1. If the authors have adequately addressed your comments raised in a previous round of review and you feel that this manuscript is now acceptable for publication, you may indicate that here to bypass the “Comments to the Author” section, enter your conflict of interest statement in the “Confidential to Editor” section, and submit your "Accept" recommendation.

Reviewer #1: All comments have been addressed

Reviewer #2: All comments have been addressed

2. Is the manuscript technically sound, and do the data support the conclusions?

Reviewer #1: Yes

Reviewer #2: Yes

3. Has the statistical analysis been performed appropriately and rigorously? 

Reviewer #1: Yes

Reviewer #2: Yes

4. Have the authors made all data underlying the findings in their manuscript fully available?

Reviewer #1: Yes

Reviewer #2: Yes

5. Is the manuscript presented in an intelligible fashion and written in standard English?

Reviewer #1: Yes

Reviewer #2: Yes

6. Review Comments to the Author

Reviewer #1: (No Response)

Reviewer #2: All suggestions were flawlessly implemented. The manuscript can be accepted for publication. The editorial board should correct two grammatical errors:

p.16, l.13: Correct would be "the numberS (plural!!) of impacts range from ... to ..."

p.21, l.7: This change is incorrect (i.e. "wrote"). The previous "written" is correct: "studies [...] written in English were included"

7. PLOS authors have the option to publish the peer review history of their article (what does this mean?). If published, this will include your full peer review and any attached files.

Reviewer #1: No

Reviewer #2: **Yes: **Philip X. Fuchs

---

## [Editor Report · Acceptance letter]

3 Aug 2020

PONE-D-19-33961R3 

Accelerometry as a method for external workload monitoring in invasion team sports. A systematic review 

Dear Dr. Gómez-Carmona:

I'm pleased to inform you that your manuscript has been deemed suitable for publication in PLOS ONE. Congratulations! Your manuscript is now with our production department. 

Kind regards, 

on behalf of

Dr. Cristina Cortis 

Academic Editor

PLOS ONE